# Ultraviolet-Sensitive Properties of Graphene Nanofriction

**DOI:** 10.3390/nano12244462

**Published:** 2022-12-15

**Authors:** Gaolong Dong, Shuyang Ding, Yitian Peng

**Affiliations:** College of Mechanical Engineering, Donghua University, Shanghai 201620, China

**Keywords:** graphene, ultraviolet light, irradiation modification, defects, friction aging

## Abstract

The friction characteristics of two-dimensional materials in the ultraviolet (UV) radiation environment are important to the reliability of two-dimensional material nano-structures of space equipment. A novel mechanism of UV light-sensitive nano-friction on graphene was proposed by ultraviolet vacuum irradiation modification using an atomic force microscope (AFM). The surface roughness, adhesion force, and friction of graphene were gradually reduced over a time of irradiation below 3 min. UV185 passes through graphene and causes photochemical reactions between its bottom layer and Si/SiO_2_ substrate, resulting in hydroxyl, carboxyl, and silanol suspension bonds and sp3-like bonds, which enhances the binding energy of graphene on the substrate and inhibits the out-of-plane deformation resulting in roughness and friction reduction. However, as the irradiation time increased to 5 min, the friction force increased rapidly with the aging effect and the breakdown of sp3-like bonds between the graphene–substrate interface. This study presents a new method of controlling nanofriction on graphene based on UV irradiation-sensitive posterities in vacuum conditions, which is essential to the application of two-dimensional materials in aerospace equipment, to improve anti-aging properties and wear reduction.

## 1. Research Background

Electronic components and microelectromechanical systems (MEMS) are important components in the field of aerospace equipment. In the complex space environment, the stability and reliability of their hardware functions are challenged to the extreme. In a vacuum environment, wear and aging cause rapid MEMS actuator failure [1]. Graphene, which has excellent mechanical and electrical properties, has resulted in many technological breakthroughs and applications in the field of aerospace equipment. Research shows that the high gas permeability resistance of graphene can effectively enhance the corrosion resistance of aerospace materials. Excellent electrical conductivity and ultra-low friction characteristics effectively reduce the energy consumption of power components [2,3]. In recent years, the electrical and frictional properties of graphene under a high vacuum radiation environment have attracted extensive attention, which is expected to promote the further application of graphene composites in the aerospace field.

The frictional properties of functionalized modified graphene have expanded its application in various fields. Among them, after plasma modification and chemical modification, light irradiation modification [4,5] is another highly efficient, stable, and clean controllable modification method, which optimizes the shortcomings of damage to the physical structure of graphene after plasma treatment [6] and surface reagent residue after chemical treatment [7]. The graphene field effect transistor obtained via UV light irradiation under vacuum condition can maintain its physical structure completely [8], but under different gas or liquid environments, UV vacuum irradiation causes graphene N-type doping and P-type doping [9,10]. Research has shown that light irradiation can produce defects on the surface of graphene and also cause photochemical reactions between graphene and the substrate layer, which produce changes to the electrical and mechanical properties of graphene, but the mechanism and change pattern are still unclear, so UV vacuum irradiation is expected to be a new method to modulate the frictional properties of graphene.

In this paper, the graphene samples were modified via UV vacuum irradiation with different durations. AFM was used to study the effect of UV vacuum irradiation time on the tribological properties of graphene, and to explore the mechanism and method of UV vacuum irradiation regulation on the nanofriction properties of graphene. This study contributes to the further application of graphene in of aerospace field.

## 2. Experimental

### 2.1. Preparation of Graphene Samples

In this study, dry oxide N-doped silicon wafers (coated with 300 nm SiO_2_) were used as the substrate. First, the silicon substrate was ultrasonically treated for 10~15 min by acetone, ethanol, and deionized water to remove surface impurities, and then placed in a nitrogen environment for natural drying. After that, the graphene was stripped from the single-crystal graphite using a mechanical stripping method and transferred to SiO_2_/Si substrate, and then transferred to a drying oven at 80 °C for 1 min to weaken the van der Waals force between the graphene sheets and to improve the quality of the mechanical stripping method [11]. The above sample preparation process was completed in a thousand-level ultra-clean working environment. Figure 1a shows the optical microscope image of graphene on SiO_2_/Si substrate, and the corresponding topography of the red box region measured by AFM tapping mode is shown in Figure 1b. Figure 1c shows the height of the red transect position in Figure 1b. Figure 1d corresponds to the surface morphology of graphene with a thickness of 0.69 nm, 1 μm × 1 μm, and the surface roughness of this area was determined as 148.520 pm.

The theoretical thickness of a monolayer of graphene is 0.34 nm. However, due to the influence of factors such as the quality of mechanically peeled single-crystal graphite and the environment at the time of transfer, it is difficult for graphene nanosheets to be closely bonded with the substrate [12,13]. Generally, the thickness of single-layer graphene on SiO_2_ substrate appears to be 0.6–1.2 nm [14]. The sample in Figure 1 has a height of 0.69 nm and can be regarded as monolayer graphene. The thickness and roughness of the remaining graphene samples were determined using the same method, and graphene with the same surface thickness and similar roughness was selected for subsequent experiments.

UV vacuum irradiation uses a BZS250GF-TS type UV Ozone Cleaner and low-pressure mercury lamp (250 W) as the radiation source. The working parameters are shown in Table 1. Before the experiment, the graphene samples were placed in a vacuum tube with a spacing of 10 cm between the samples and the lamp, and all of them were exposed to the lamp. At the same time, the air pressure in the tube was pumped to 10^−2^ MPa through the air pump, as shown in Figure 2. Irradiation was carried out at room temperature. Ten groups of graphene samples with similar thickness and roughness were irradiated with UV vacuum at different times to study the effects of different irradiation times on the tribological properties of graphene.

### 2.2. Experimental Method of Friction

An MFP-3D atomic force microscope (AFM) from Asylum Research was used in this study. The probe adopts a silicon probe with a tip of less than 10 nm (Multi75 Al-G Budget Sensors). The normal elastic coefficient of the probe cantilever is calibrated using the Sader thermal noise method, and the lateral sensitivity is determined using the non-contact calibration method. In order to study the effects of UV vacuum irradiation time on the tribological properties of graphene, the experiment must be carried out with the same probe under the same measuring environment conditions (temperature: 25 °C ± 2 °C, humidity: 45% ± 5% R.H).

The measurement experiment steps are as follows. First, find and determine the sample area of each experiment in the tapping mode to ensure that each experiment is in the same position. After calibrating the probe normal and lateral sensitivity of probe coefficients in contact mode, the adhesion force, friction–load relationship, and stick–slip characteristics in the 500 nm × 500 nm region were tested to determine the initial tribometric properties of graphene. A UV vacuum irradiation timing experiment was conducted, and an AFM characterization experiment was conducted immediately after irradiation. The specific measurement steps were the same as above. In the adhesive force test, 10 nN loads are taken uniformly, and the adhesive force values are taken from 5 different positions on the same sample and the average value is obtained. The morphology, roughness, adhesion force, friction force, and stick–slip characteristics of graphene under different UV vacuum irradiation durations were compared and analyzed, and the methods and mechanisms of regulating the friction characteristics of graphene were explored.

## 3. Results and Discussion

### 3.1. Tribological Characteristics under Short-Term UV Irradiation

This study defined the short-term experimental conditions at an ambient temperature of 25 °C ± 2 °C and humidity of 45% ± 5% R.H to pump the tube to vacuum (<10^−2^ MPa), and modified by irradiation within 2 min in a UV ozone cleaning machine. Figure 3a shows the optical microscope images of the same graphene sample after UV vacuum irradiation for 5 s, 10 s, 30 s, and 60 s. There was a slight change in the color of graphene as the UV vacuum irradiation time increased. The AFM morphology of corresponding positions is shown in Figure 3b, which directly shows the influence of UV vacuum irradiation time on the height of graphene. Figure 3c shows the friction diagram of single-layer graphene at a 10 nN load in Figure 3b. It can be obviously observed that the friction gradually decreases with the increase in UV vacuum irradiation time. Figure 3d shows the height changes of graphene samples after UV vacuum irradiation at different times. It is observed that the sample thickness decreases with the increase in UV irradiation time, the specific height can see the Appendix A. This reflects a significant decrease in the interlayer distance between graphene and the substrate.

As the main factor affecting friction, surface roughness can be used as one of the criteria to determine the change in graphene properties before and after UV vacuum irradiation. Figure 3e shows that with the increase in short-term UV vacuum irradiation time, the surface roughness of graphene shows a decreasing trend.

The roughness of graphene is closely related to the external environmental factors and operation methods in the transfer process. Corrugations caused by the transfer and complex interlayer environment cause the surface roughness of graphene after transfer to increase [11,13]. It is known that there are silanol groups on the surface of SiO_2_, which have a strong adsorption effect on water molecules. When mechanically peeled graphene was transferred to Si/SiO_2_ substrate, there was a small amount of trapped water in the interface between graphene and SiO_2_ [8]. Due to the extremely high gas permeability resistance of graphene, O_2_ molecules are difficult to clean in vacuum, and there could be some residue in the layers. UV irradiation of SiO_2_ samples in water leads to the increase in hydroxyl groups on their surface, and the experimental results say that the O-H bond in water molecules can be photolysis [15]. With UV vacuum irradiation, the trapped water and silanol groups between the graphene–substrate interface are dissociated to form hydrogen radicals, hydroxyl radicals, and siloxyl radicals. These free radicals tend to react with graphene lattice, forming a kind of sp3 structure [8], which can strengthen the combination between the graphene and basal, reducing graphene layer spacing and roughness.

The frictional properties of graphene were investigated after UV irradiation. Figure 4 shows the load reduction friction and adhesion of graphene nanosheets of three thicknesses measured after UV vacuum treatment at different times. The data show that the friction on graphene surfaces of different thicknesses decreases with the increase in UV vacuum irradiation time, as shown in Figure 4a–c. It was found that the frictional force of graphene decreased significantly after UV vacuum irradiation, and the decreasing frictional force trend diminished with the increase in thickness, SM Appendix A also confirms this phenomenon. This is mainly due to two reasons: first, the layering effect in the frictional properties of graphene—the frictional force decreases with the increase in the number of graphene layers. Second, the light transmission of thin-layer graphene is better than that of thick-layer graphene [16], the trapped water between the graphene and substrate interface is more likely to dissociate and form sp3-like bonds to attenuate the out-of-plane deformation of graphene. The interaction between graphene and substrate reflects the adhesion between the probe tip and graphene. The stronger the substrate is bound, the weaker the adhesion will be; conversely, a weaker binding strength produces greater adhesion [12]. Figure 4d clearly shows that the adhesion between the tip and graphene decreased with the UV irradiation time; when treated with UV vacuum irradiation for 60 s, the adhesion of graphene under the same load reduced by 4.05 nN, which is about four times less than the initial adhesion strength, which is caused by the surface modified properties and the bonding strength between graphene and basal surface.

Figure 5 shows the curves of graphene friction with different thicknesses as a function of UV vacuum irradiation time under a load of 10 nN. As shown in Figure 5a, the friction force of graphene film with a thickness of 0.69 nm decreases from 0.787 nN to 0.469 nN. In Figure 5b,c, the friction of graphene films with a thickness of 0.89 nm and 2.82 nm also decreased by 0.236 nN and 0.111 nN. Compared with 2.82 nm graphene films, the friction of 0.69 nm and 0.89 nm graphene films decreased significantly. According to the principle of friction of micro-nano scales, the friction is associated with the direct contact area of two objects. Because of the small stiffness, there are a lot of ups and downs on the graphene surface after mechanical transfer due to the folding effect, which could cause a larger frictional force at the nanoscale when subjected to the shear action of the needle tip, due to the folding effect. Lee et al. [17] found that when the needle tip slid on the loose graphene, folds would appear in the needle tip and increase the contact area between the graphene and the needle tip. Due to the low out-of-plane stiffness of graphene and the weak binding effect between the graphene and the substrate, part of the graphene is separated from the substrate and wrapped around the tip of the needle, thus increasing the real contact area between graphene and the tip, resulting in increased friction. In contrast, the strong interaction between the substrate and the tip reduces the out-of-plane deformation of the graphene, reducing the contact area between the tip and the graphene, and ultimately reducing the friction force. For example, the friction behavior of graphene deposited on mica was found to be lower than that of graphene deposited on silicon substrate [18]. In this study, it was found that short-term UV vacuum irradiation can reduce the graphene layer spacing and roughness and obtain a stable and low friction effect. The experiment further shows that the bonding strength between graphene and substrate has an important effect on the friction properties of graphene.

In order to reveal the effect of UV vacuum irradiation on the bonding strength between graphene and SiO_2_ substrate, atomic-scale stick–slip tests were performed on thin layers of graphene under the same load (10 nN) and scanning speed. The surface of thin graphene shows friction enhancement due to the folding phenomenon [12,18]. This experiment shows significant friction force enhancement in Figure 6a–d. However, with the UV vacuum treatment time, the binding effect between graphene and the substrate interface increases. The evolution of fold and contact quality are significantly inhibited and constrained [12], then the friction enhancement effect is gradually weakened, as shown in Figure 6e,f. The friction regularity of the atomic scale indicates that UV vacuum irradiation could affect the out-of-plane deformation of graphene. After a certain time, the thin layer of graphene also shows the friction behavior of the thick layer of graphene, and the friction enhancement phenomenon disappears, which reflects the gradual strengthening of the binding between the substrate and graphene from the side. This is mainly attributed to the extremely high light transmissibility of graphene. A large part of the energy of 185 nm ultraviolet rays passes through graphene, photolysis water on the surface of SiO_2_, forming several suspended bonds, such as hydroxyl and silyl, which increase the surface energy of the substrate and increase with the increase in the treatment time. Moreover, sp3-like bonds are generated in the vertical direction of the interface. Thus, the interface binding between graphene and the substrate is enhanced, and the ability of graphene to deform out of the plane is inhibited, which has a close impact on the surface friction properties of graphene. Other studies have shown that friction force is closely related to the contact mass between two objects, which indicates that friction characteristics can be regulated by changing the contact mass without changing the structural characteristics of the object surface [19,20].

Figure 7 illustrates the transfer of mechanically peeled graphene to the upper surface of the substrate with large fluctuations and trapped water molecules and silanol groups remaining in the interlayer. The fluctuation of the graphene surface is caused by environmental factors, transfer means, and the properties of graphene itself. Graphene has a high specific surface area, so it is easy to curl and fold. Therefore, graphene can wrap the tip of a needle well and increase the contact area.
(1)$$H_{2}O\overset{hv(185\;\mathrm{nm})}{\to }\mathrm{HO}•+H•\;\varphi_{185}\left(\mathrm{OH}•\right)=0.33$$
$$H_{2}O\overset{hv(185\;\mathrm{nm})}{\to }\mathrm{HO}•+H^{+}+e_{aq}{}^{-}\;\varphi_{185}\left(+e_{aq}{}^{-}\right)=0.045$$

The hydration electron: the REDOX potential is −2.9 V, and φ represents the quantum yield of each reaction [21].
(2)$$O_{2}\overset{hv(185\;\mathrm{nm})}{\to }2O\;O+O_{2}+M\overset{}{\to }O_{3}+M$$
$$O_{3}\left(\mathrm{gaseous}\right)\overset{hv(254\;\mathrm{nm})}{\to }O+O_{2}\;\varphi_{254}\left(-O_{3}\right)=0.099$$

In Formula (2), M stands for the remaining energy from other molecules in the environment, and the quantum yield of O_3_ formation is 0.009.

Graphene has extremely high resistance to gas permeability and light transmission. So, in UV vacuum irradiation modification, when a portion of UV light is irradiated through graphene on the substrate, the trapped water molecules and oxygen atoms between the interface of the graphene and substrate undergo photochemical reactions (see Formulas (1) and (2)), such as hydrogen bonding, hydroxyl. It is the existence of this part of the hanging key that enhances the interaction between the graphene and basement. To some extent, the folding state of the transferred graphene is slowed down (as shown in Figure 7b, and the properties of the graphene’s own low out-of-plane bending stiffness are changed. As a result, when the tip slips on the surface, the actual contact mass (the actual contact area and the overall structure of the extensibility) is reduced, so the friction reduction phenomenon occurs.

### 3.2. Frictional Aging under Long-Term UV Vacuum Irradiation

Long-time UV vacuum irradiation provides graphene more energy absorption. Under the excitation of high energy, graphene produces partial defects and even layer reduction. As a kind of high-energy ray, ultraviolet ray has enough energy to cut the chemical bonds in most substances. When the ground state of a substance absorbs photons, it forms an excited state, which is followed by photophysical and photochemical reactions, resulting in photoionization, intramolecular rearrangement, free radical formation, and other phenomena [22]. Although graphene has a lot of UV light penetration, there is still a part of the energy that is absorbed, which causes the bond energy of the C-C bond of graphene to decrease and even dissociate to form vacancy defects. This has a crucial effect on the mechanical properties of the graphene. It can be clearly observed from Figure 8a that after a long period of UV vacuum irradiation, the C-C bond on the surface of the graphene film breaks, forming a large number of vacancy defects, which is the main cause of friction enhancement [23,24,25]. Figure 8b shows the morphology of the graphene film irradiated by a UV vacuum for a long time after the surface was scratched by the probe under 40 nN load, showing that the strength of the carbon–carbon bond and surface strength of graphene decreased after UV vacuum irradiation for a long time, details of the experiment can see the SM Appendix A. Figure 8 represents the schematic diagram of graphene surface defects after UV vacuum irradiation. The study of B.K. Mund [26] et al. found that mechanically transferred single- or few-layer graphene undergoes varying degrees of oxidation under UV vacuum irradiation in a humid environment, but for multilayer graphene, oxidation under UV irradiation occurs only near the substrate layer due to photolysis of trapped water between the graphene–substrate interface under UV irradiation.

Figure 9a shows the friction comparison of graphene with 1 nm thickness after UV vacuum irradiation for 3 min, 5 min, 30 min, 60 min, and 120 min. It obviously shows that the surface friction increases with the extension of UV vacuum irradiation time. Figure 9b shows the comparison of the friction of graphene with the same thickness after UV vacuum irradiation at different times under a load of 20 nN. The red area can be considered the short-time UV vacuum irradiation that reduces friction, while the blue area shows the aging of the irradiated graphene and the increase in friction. The combination of the two can achieve bidirectional regulation of graphene friction by adjusting the length of UV vacuum irradiation time. The change in friction properties of 1 nm thickness graphene UV vacuum irradiation after 1 h can be see the SM Appendix A.

Vacuum irradiation is the aging of graphene, that is, the formation of surface defects of graphene after UV vacuum irradiation, as shown in Figure 8c. Firstly, the mechanism of defect formation after ultraviolet irradiation is considered an optical transition from the π to π* band. According to the linear momentum and energy transfer, the constant absorption of graphene to visible light is 2.3% [27,28]. In the ultraviolet region, the absorption is no longer linearly increased with the dispersion. The absorption is at a maximum at a photon energy of 4.62 eV, which is caused by the optical transition at point M in the graphene Brillouin region [29]. Therefore, the electron transition from π to π* band of graphene absorbs UV light energy, and the transformation from σ to σ* may also form defects. It has also been reported that the conversion from σ to σ* occurs when the photon energy is 8.3 eV in exciton resonance [30]. Although the photon intensity of a mercury lamp decreases with the increase in photon energy, G. Yoshikawa et al. [31] observed photon emissions with energy greater than 8.3 eV. Therefore, the transition from σ to σ* is excited by ultraviolet irradiation of a low-pressure mercury lamp, and the excited states formed by the σ-σ* transition and π-π* transition can lead to the weakness of the carbon–carbon bond and lead to the dissociation and recombination of the carbon–carbon bond, resulting in the formation of structural defects in graphene. The three-body potential can describe substances containing covalent bonds. The Tersoff–Brenner [32] potential has been widely used in the study of friction properties of graphene and can clearly describe the interaction of C-C covalent bonds in graphene and the interaction of Si-Si covalent bonds in the probe. The expression of the potential function is as follows:(3)$$E_{\mathrm{tersoff}\;}=\sum_{i}\sum_{j(i>j)}f_{c}(r_{ij})\{V_{R}(r_{ij})-b_{ij}V_{A}(r_{ij})\}$$
where *E*_tersoff_ is the potential energy between the graphene substrate and the silicon probe, $f_{c}(r_{i}{}_{j})$ is the cut-off function, *V_R_* is the repulsion and separation, *V_A_* is the attractive part, *r_ij_* is the distance between atom *i* and atom *j* in the graphene substrate or silicon probe, and *b_ij_* is the measure of bond order [33]. There is a van der Waals force between the probe and the substrate, so the interaction between C-Si adopts the Lennard-Jones potential, which can be expressed as:(4)$$E_{LJ}=\sum_{i}\sum_{j(i>j)}4\varepsilon_{C-Si}\left[{\left(\frac{\sigma_{C-Si}}{r_{ij}}\right)}^{12}-{\left(\frac{\sigma_{C-Si}}{r_{ij}}\right)}^{6}\right]$$
where *E_LJ_* is the van der Waals potential between graphene and probe, the potential well depth = 8.0909 meV [31], the equilibrium constant = 0.3326 nm, *r_ij_* is the distance between graphene carbon atom *i* and probe silicon atom *j*.

The site where graphene defects are formed has higher activity, which increases the interaction potential between the needle tip and the graphene interface. As a result, the needle tip needs more energy to overcome the high barrier when passing through the defect [34,35], resulting in greater friction force. At the same time, the appearance of defects on the graphene surface is random, and the friction can be increased stably only when the number of defects is certain [36,37]. Therefore, the defects caused by short-term UV vacuum irradiation are not enough to increase the friction of graphene, but when UV irradiation is afforded a certain amount of time, the number of defects increases sharply, resulting in a rapid increase in friction. In addition, the emergence of a large number of vacancy defects increase the out-of-plane flexibility of graphene, which also generate greater friction by increasing the out-of-plane deformation. Therefore, long-term UV vacuum irradiation produces a large number of structural defects on the graphene surface, thus regulating the nonlinear increase in graphene friction.

## 4. Conclusions

In this study, it was found that the height and surface roughness of thin graphene tended to decrease after short-term UV vacuum irradiation. Then, the friction also decreased with the increase in UV vacuum irradiation time, and the friction enhancement phenomenon in stick–slip behavior was significantly inhibited. The reason is that UV185 passes through graphene and causes photochemical reactions between its bottom layer and Si/SiO_2_ substrate, resulting in hydroxyl, carboxyl, and silanol suspension bonds and sp3-like bonds, which enhances the binding energy of graphene on the substrate, inhibits out-of-plane deformation, reduces the folding effect, and leads to the reduced roughness and improved friction properties of graphene; yet, after long-term UV vacuum irradiation, the surface morphology of the graphene produced an irregular mesh, and the surface was easily destroyed under shear force. At the same time, the surface friction increased obviously with the increase in UV vacuum irradiation time. Because of the strong absorption effect of graphene on UV254, the accumulation of high energy leads to the reduction in the bond energy of the graphene C-C bond until dissociation recombination occurs, resulting in a large number of vacancy defects on the surface of graphene, leading to surface aging and friction increasing.

The results of this research present a new method of UV vacuum irradiation modification of graphene. The bidirectional regulation of the tribological properties of graphene was achieved through UV vacuum irradiation. There are more potential oxidation mechanisms under UV irradiation in vacuum/air and more applications of this UV light-sensitive control method in graphene nanofriction as a carrier and lubricant for micro and nanosensors in aerospace equipment.

## Figures and Tables

**Figure 1 nanomaterials-12-04462-f001:**
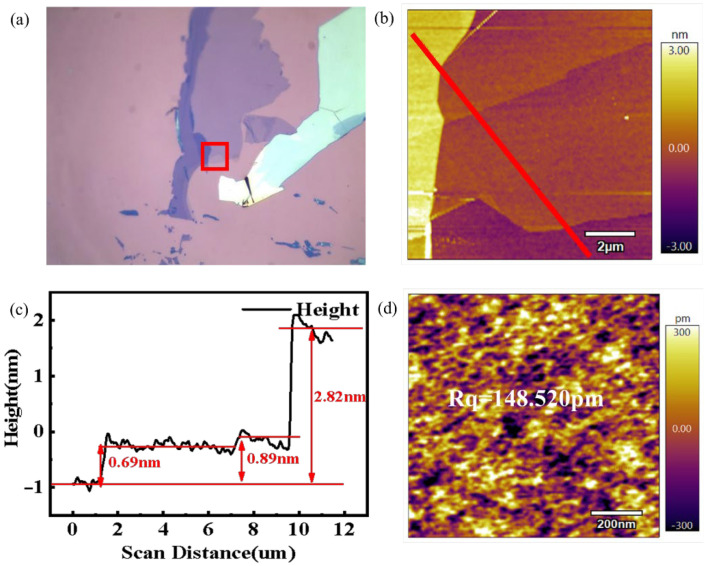
Graphene nanosheets of different thicknesses. (**a**) Optical microscopy, (**b**) AFM morphology, (**c**) height of the corresponding transect in the morphology, (**d**) 0.69 nm microstructure and roughness of graphene with different thickness. The red line in (**a**) represents the actual position of the topography, and the red line in (**b**) represents the corresponding cross−section of the height map.

**Figure 2 nanomaterials-12-04462-f002:**
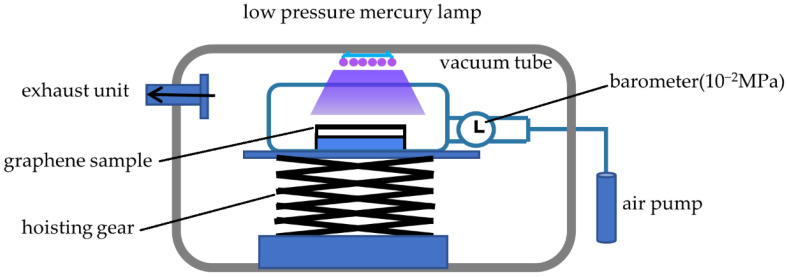
Experimental equipment model.

**Figure 3 nanomaterials-12-04462-f003:**
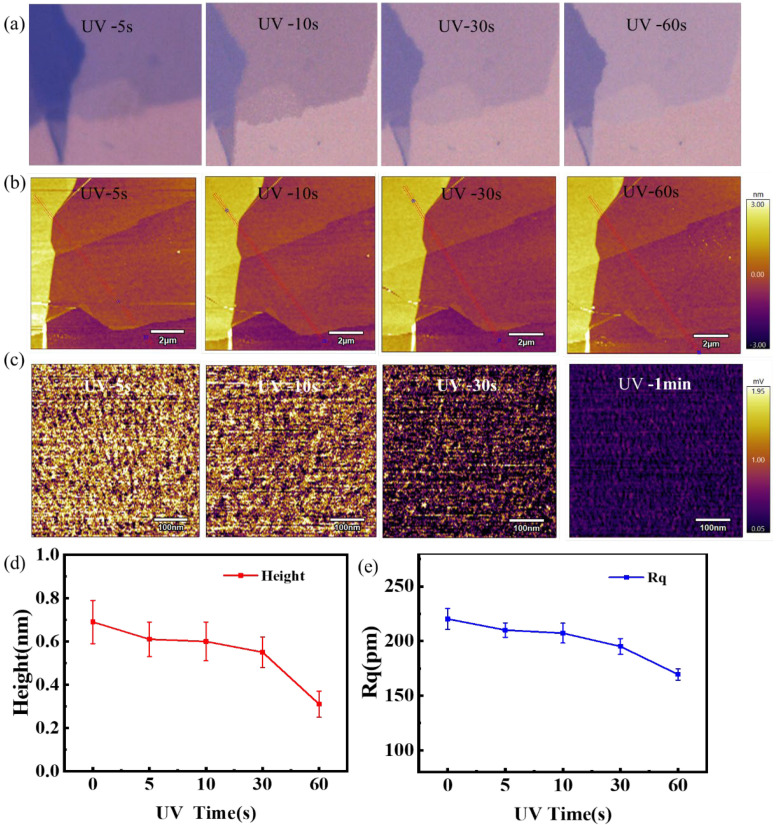
(**a**) Light microscope image of a graphene sample after UV irradiation in vacuum; (**b**) AFM topography map of the corresponding region to (**a**); (**c**) represents the surface sweep friction diagram (10 nN load) of the corresponding single-layer region in (**b**); (**d**,**e**) represent the variation trend of the height and roughness of monolayer graphene samples with UV vacuum irradiation time, respectively.

**Figure 4 nanomaterials-12-04462-f004:**
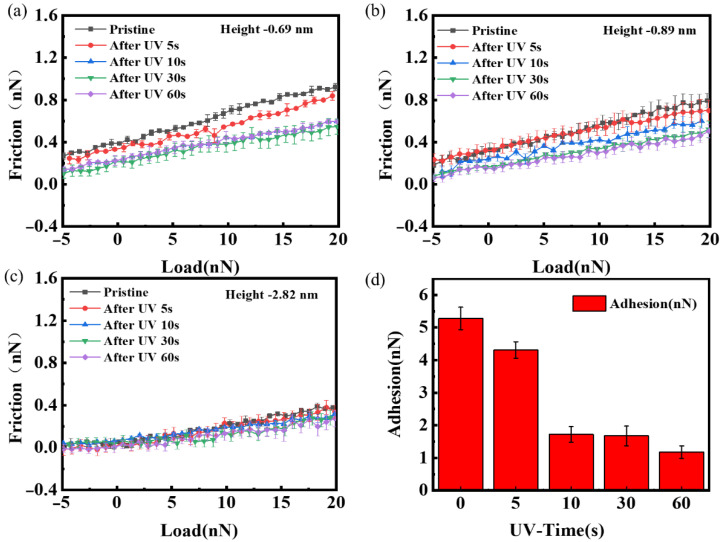
(**a**–**c**) The relationship between friction force and load of different graphene nanosheets in the UV vacuum irradiation time range of 0 s to 60 s. (**d**) Adhesion of graphene after UV vacuum treatment at different times (10 nN load).

**Figure 5 nanomaterials-12-04462-f005:**
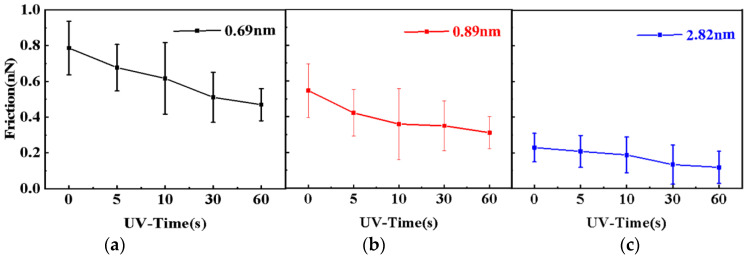
The relationship between friction force of graphene with different thicknesses and UV vacuum irradiation time: (**a**) 0.69 nm, (**b**) 0.89 nm, (**c**) 2.82 nm.

**Figure 6 nanomaterials-12-04462-f006:**
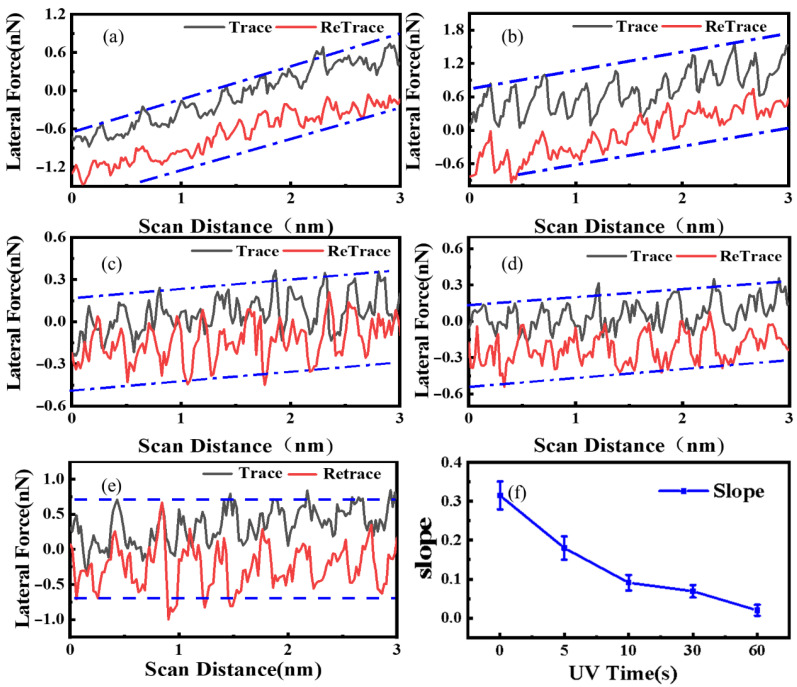
The friction force of graphene treated with UV vacuum at different times varies: (**a**) 0 s, (**b**) 5 s, (**c**) 10 s, (**d**) 30 s, (**e**) 60 s. (**f**) The values present the blue lines in (**a**–**e**) based on the least squares fitting method.

**Figure 7 nanomaterials-12-04462-f007:**
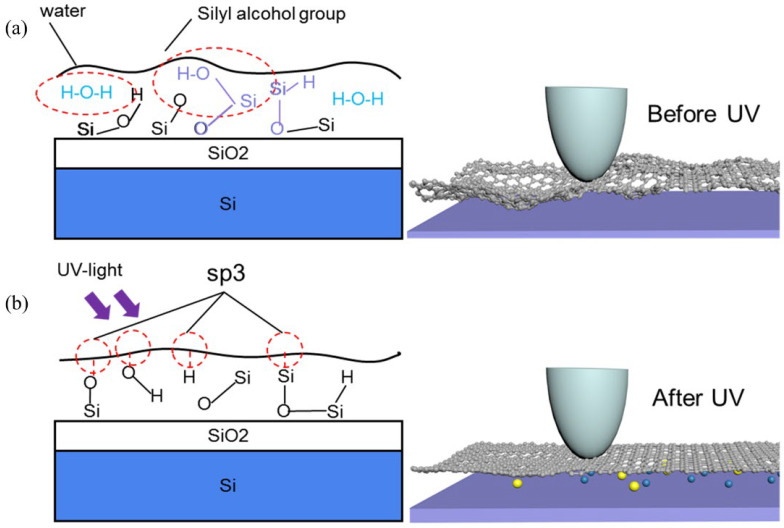
(**a**) Schematic diagram of hanging bond changes between graphene and substrate before and after UV irradiation. (**b**) Schematic diagram of friction after UV vacuum treatment of graphene.

**Figure 8 nanomaterials-12-04462-f008:**
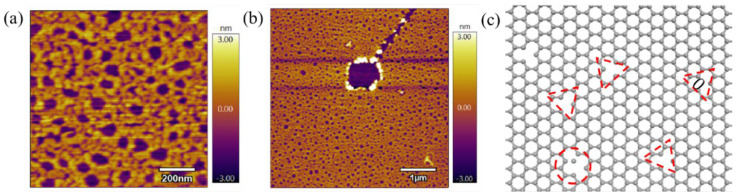
AFM morphology of graphene after long-term UV vacuum irradiation: (**a**) surface microscopic defects; (**b**) topography after friction under 40 nN load; (**c**) schematic diagram of defects after UV vacuum irradiation.

**Figure 9 nanomaterials-12-04462-f009:**
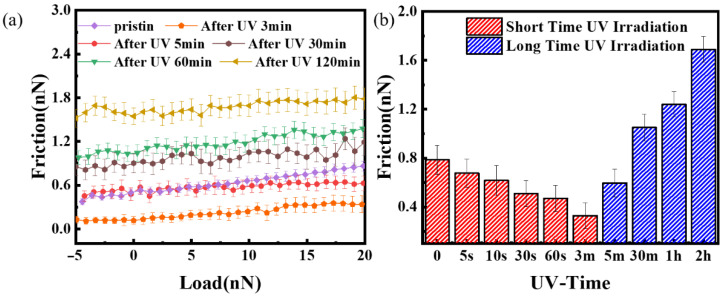
(**a**) Friction comparison of 1 nm thick graphene after UV long-term vacuum irradiation; (**b**) friction comparison of graphene with the same thickness after UV vacuum irradiation at different times under 20 nN loads. The red area can be classified as short-time irradiation reduction in friction, while the blue area is long-term irradiation aging.

**Table 1 nanomaterials-12-04462-t001:** Experimental parameters of BZS250GF-TS UV radiometer.

Tube Type	Special Lattice Lamp Tube (Imported Quartz Material), 250 W
Work spectrum	185 nm + 254 nm
Effective power density	26~30 mW/cm^2^
Distance between sample and lamp tube	100 mm
Ozone exhaust system	III

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
