# Peer review of "Ultraviolet-Sensitive Properties of Graphene Nanofriction"

_nanomaterials, 2022, doi:10.3390/nano12244462_

Round 1

Reviewer 1 Report

Obtained results are interesting, but there are a lot of unreadable sentences and mistakes. 

The main question is: Why oxidation effect is not considered? The pressure of 10-2 MPa is not a high vacuum, there are a lot of residual air, so ionization of oxygen and formation of ozone under UV irradiation is possible. 

Another important question is: How can this effect help to  use graphene in aerospace applications mentioned in Introduction?

Other questions and remarks are listed below:

1) line 25: "In a vacuum environment, MEMS actuators will accelerate due to wear and aging failure." What does it mean? Actuators will accelerate, i.e. increase a velocity of something?

2) line 60:  "... van der Waal forces ...", van der Waals? Probably misprint.

3) line 60-61: " ... 80℃ for 1 minute ... to improve the dispersion of the graphene." What is the dispersion of the graphene in this case? Width at half maxima, or separation of light into colors, or what?

4) line 77: "Therefore, it is generally considered that mechanically peeled graphene within the thickness of 1 nm is a monolayer[8]." The authors claim that it is "generally considered", does it mean that measured thickness of graphene monolayer  is always of 1 nm, and it's impossible to observe 0.34-nm-thick film? If not, the question is in what cases we can say, that 1 nm corresponds to single layer?

 5) line 84: The authors call their setup "BZS250GF-TS type UV radiometer". Radiometer is the device used for the intensity measurements. Is it really a radiometer or ozone cleaner produced by Hwotech? Please provide more details.

6) line 122 and figure 3a): "... the color of graphene becomes lighter and lighter with the increase of UV vacuum irradiation time." To make sure that the color of graphene really becomes lighter, but not the settings of the camera was changed, the substrate should have the same color in all figures. All areas in the leftmost figure in figure 3a) are darker then in other figures, so it looks like artifact, caused by camera settings.

7) lines 127-130: "Figure 3(d) shows the height changes of four groups of graphene samples with similar heights after UV vacuum irradiation at different times. It is observed that the sample thickness decreases with the increase of UV irradiation time, gradually approaching the theoretical  monolayer thickness (0.36 nm)." Where it is shown in Figure 3(d)?  I see the height of about 1.1 nm, but not 0.36 nm, slightly decreasing with irradiation time. Moreover, there are two curves labeled "before UV" and "after UV", nothing about four samples. What does it mean? The height vs irradiation time for the sample before UV is meaningless, since irradiation time before UV is zero.

8)  line 135-137: "Figure 3(e) shows that with the increase of short-term UV vacuum irradiation time, the surface roughness of graphene has an obvious downward trend, and is inversely proportional to the UV vacuum irradiation time." Similarly to previous point it is not clear what two curves are shown in figure, and how the roughness before UV may depend on irradiation time. The statement that the roughness is "inversely proportional to the UV vacuum irradiation time" is too strong without confirmation by corresponding curve fit. 

9) lines 152-153:"UV irradiation of SiO2 samples in water leads to the increase of hydroxyl groups on their surface, ..."  What does increase of the hydroxyl groups mean? Which value increases?

10) lines 162-163: "The data show that the friction on graphene surfaces of different thicknesses decreases with the increase of UV vacuum irradiation time, as shown in Figures 4(a)-(e)." These figures show that the friction increases with load, but the dependence on irradiation time is not shown. Each figure presents one irradiation time, so to see the dependence of the friction on irradiation time I should take some data from different figures and plot it by myself. Actually, it is shown in figure 5.

11) line 168: " ...the friction decreases with the increase of layer." What does it mean? Is "layer" a value, which may increase?

12) line 168-169: "the light penetration of thin-layer graphene is better than that of thick-layer graphene, and the out-of-plane deformation is more easily affected by the interlayer forces." Please estimate the absorption values for these thicknesses and compare to each other. How the light penetration is connected with "out-of-plane deformation" and "interlayer forces"?

13) lines 192-194: "..., mechanical transfer graphene because in the process of transfer from all directions of force, the surface there are a lot of ups and downs, and stiffness is small, when from the reaction between the needlepoint, due to folding effect caused by the larger frictional forces." Some words are probably missed here, making this sentence unreadable.

14) line 220: "The friction laws of atomic scale indicate that UV vacuum irradiation could affect the out-of-plane deformation of graphene." What laws? Please specify with appropriate references.

15) lines 245-249: Some values called "phi" are presented in equations, but their meaning is not explained.

16)line 269: "Although graphene is very high for UV light through sex, ..." Through what?

Author Response

We would like to thank the reviewers for a thorough review and useful comments. Those comments were helpful for us to revise and improve our paper, and all changes suggested by the reviewer have been incorporated. We have done our best the respond to all the suggestions/comments of reviewers and all the replies are listed in the attached documents. If there are further changes to be made, we will be happy to comply.

Reviewer 2 Report

Dear Authors,

The manuscript submitted for review, titled. "Ultraviolet-sensitive properties of graphene nano-friction" by Gaolong Dong, Shuyang Ding and Yitian Peng concerns study the effect of UV vacuum irradiation time on the tribological properties of graphene and determine the mechanism and method of UV vacuum irradiation regulation on the nanofriction properties of graphene. The manuscript is interesting and addresses current issues related to aspects of its use in aircraft. It is written in accordance with the requirements, and its structure is appropriate, the sources cited are up-to-date and relevant. It should be concluded that the content of the presented work corresponds to the topic specified in its title. The manuscript is written correctly, addresses topical issues and can be useful for those dealing with the above issues and application of graphene in the field of aerospace to improve anti-aging properties and wear reduction.

The proposed methodology is modern and correct, described in a clear and concrete manner, leading to the conclusion that the studied parameters, i.e. the surface roughness, adhesion force, and friction of graphene were gradually reduced as the time of irradiation. The paper has an adequate and appropriate number of literature sources, although undoubtedly many more interesting items can be found.

I have no major objections to the content of the manuscript, however, before publication, one could consider whether to make a more in-depth and scientific analysis of the color change of graphene, which becomes lighter and lighter with the increase of UV vacuum irradiation time, for example, by performing tests with a calorimeter or spectrophotometer.

Once this aspect is included in the content of the manuscript, it can be published in the journal Nanomaterials.

Best regards,

Author Response

We would like to thank the reviewers for a thorough review and useful comments. Those comments were helpful for us to revise and improve our paper, and all changes suggested by the reviewer have been incorporated. We have done our best the respond to all the suggestions/comments of reviewers. If there are further changes to be made, we will be happy to comply.

Round 2

Reviewer 1 Report

Dear Authors!

I still see a lot of issues in English language and styling.  Some questions and remarks are listed below.

1) lines 43-47: "Research has shown that light irradiation graphene can be formed on the surface of graphene defectss, it will also produce a photochemical reaction between graphene and the base layer, the electrical and mechanical properties were changed, but its mechanism and the change rule are not yet clear, therefore, vacuum UV irradiation is expected to become a kind of brand-new graphene used to control the friction characteristics."

It means that the defects have some surface and light irradiation graphene (what is this?) can be formed on this surface. Somehow it will produce photochemical reaction in future. Electrical and mechanical properties were changed, so it happened in the past. What is the reason of this change? Photochemical reaction that will take place in future? And why vacuum UV irradiation will be a kind of brand-new graphene? Irradiation is a process, but not a material.

2) lines 86, 93, 101, etc.: "UV vacuum irradiation" It looks like the samples were irradiated with vacuum, and the color of this vacuum was ultraviolet. Does it mean irradiation with vacuum ultraviolet light, or irradiation with UV light under vacuum conditions? Please specify.

3) lines 165-169: "... with the increase in thickness, the decreasing trend of friction decreases ... the light penetration of thin-layer graphene is better than that of thick-layer graphene..." 

Why the thickness of graphene is not the same, if it is a monolayer? The thickness of one monolayer is constant, so either the samples with different thicknesses are not monolayers, or the light penetration is the same for all samples.

4) line 185: "Figure 5 shows the curves of graphite friction with different thicknesses ..." Graphite or graphene? 

Author Response

Dear Reviewers

We would like to thank the reviewers for a thorough review and useful comments. These comments are helpful for us to revise and improve our paper, and all the changes suggested by the reviewer have been taken and carefully revised. The English language has been edited by MDPI again. If there are further changes to be made, we will be happy to comply.

best regards

all authors
